# Gradient Vaccine: Investigating and Improving Multi-task Optimization in Massively Multilingual Models

**Zirui Wang**[1,2]*, **Yulia Tsvetkov**[1], **Orhan Firat**[2], **Yuan Cao**[2]
[1]Carnegie Mellon University, [2]Google AI
{`ziruiw,ytsvetko`}@cs.cmu.edu, {`orhanf,yuancao`}@google.com

## Abstract

Massively multilingual models subsuming tens or even hundreds of languages pose great challenges to multi-task optimization. While it is a common practice to apply a language-agnostic procedure optimizing a joint multilingual task objective, how to properly characterize and take advantage of its underlying problem structure for improving optimization efficiency remains under-explored. In this paper, we attempt to peek into the black-box of multilingual optimization through the lens of loss function geometry. We find that gradient similarity measured along the optimization trajectory is an important signal, which correlates well with not only language proximity but also the overall model performance. Such observation helps us to identify a critical limitation of existing gradient-based multi-task learning methods, and thus we derive a simple and scalable optimization procedure, named Gradient Vaccine, which encourages more geometrically aligned parameter updates for close tasks. Empirically, our method obtains significant model performance gains on multilingual machine translation and XTREME benchmark tasks for multilingual language models. Our work reveals the importance of properly measuring and utilizing language proximity in multilingual optimization, and has broader implications for multi-task learning beyond multilingual modeling.

## 1 Introduction

Modern multilingual methods, such as multilingual language models (Devlin et al., 2018; Lample & Conneau, 2019; Conneau et al., 2019) and multilingual neural machine translation (NMT) (Firat et al., 2016; Johnson et al., 2017; Aharoni et al., 2019; Arivazhagan et al., 2019), have been showing success in processing tens or hundreds of languages simultaneously in a single large model. These models are appealing for two reasons: (1) Efficiency: training and deploying a single multilingual model requires much less resources than maintaining one model for each language considered, (2) Positive cross-lingual transfer: by transferring knowledge from high-resource languages (HRL), multilingual models are able to improve performance on low-resource languages (LRL) on a wide variety of tasks (Pires et al., 2019; Wu & Dredze, 2019; Siddhant et al., 2020; Hu et al., 2020).

Despite their efficacy, how to properly analyze or improve the optimization procedure of multilingual models remains under-explored. In particular, multilingual models are *multi-task learning (MTL)* (Ruder, 2017) in nature but existing literature often train them in a monolithic manner, naively using a single language-agnostic objective on the concatenated corpus of many languages. While this approach ignores task relatedness and might induce *negative interference* (Wang et al., 2020b), its optimization process also remains a black-box, muffling the interaction among different languages during training and the cross-lingual transferring mechanism.

In this work, we attempt to open the multilingual optimization black-box via the analysis of loss geometry. Specifically, we aim to answer the following questions: (1) Do typologically similar languages enjoy more similar loss geometries in the optimization process of multilingual models? (2) If so, in the joint training procedure, do more similar gradient trajectories imply less interference between tasks, hence leading to better model quality? (3) Lastly, can we deliberately encourage

---

*Work done during an internship at Google.

more geometrically aligned parameter updates to improve multi-task optimization, especially in real-world massively multilingual models that contain heavily noisy and unbalanced training data?

Towards this end, we perform a comprehensive study on massively multilingual neural machine translation tasks, where each language pair is considered as a separate task. We first study the correlation between language and loss geometry similarities, characterized by gradient similarity along the optimization trajectory. We investigate how they evolve throughout the whole training process, and glean insights on how they correlate with cross-lingual transfer and joint performance. In particular, our experiments reveal that gradient similarities across tasks correlate strongly with both language proximities and model performance, and thus we observe that typologically close languages share similar gradients that would further lead to well-aligned multilingual structure (Wu et al., 2019) and successful cross-lingual transfer. Based on these findings, we identify a major limitation of a popular multi-task learning method (Yu et al., 2020) applied in multilingual models and propose a *preemptive* method, **Gradient Vaccine**, that leverages task relatedness to set gradient similarity objectives and adaptively align task gradients to achieve such objectives. Empirically, our approach obtains significant performance gain over the standard monolithic optimization strategy and popular multi-task baselines on large-scale multilingual NMT models and multilingual language models. To the best of our knowledge, this is the first work to systematically study and improve loss geometries in multilingual optimization at scale.

## 2 INVESTIGATING MULTI-TASK OPTIMIZATION IN MASSIVELY MULTILINGUAL MODELS

While prior work have studied the effect of data (Arivazhagan et al., 2019; Wang et al., 2020a), architecture (Blackwood et al., 2018; Sachan & Neubig, 2018; Vázquez et al., 2019; Escolano et al., 2020) and scale (Huang et al., 2019b; Lepikhin et al., 2020) on multilingual models, their optimization dynamics are not well understood. We hereby perform a series of control experiments on massively multilingual NMT models to investigate how gradients interact in multilingual settings and what are their impacts on model performance, as existing work hypothesizes that gradient conflicts, defined as negative cosine similarity between gradients, can be detrimental for multi-task learning (Yu et al., 2020) and cause negative transfer (Wang et al., 2019).

### 2.1 EXPERIMENTAL SETUP

For training multilingual machine translation models, we mainly follow the setup in Arivazhagan et al. (2019). In particular, we jointly train multiple translation language pairs in a single sequence-to-sequence (seq2seq) model (Sutskever et al., 2014). We use the Transformer-Big (Vaswani et al., 2017) architecture containing 375M parameters described in (Chen et al., 2018a), where all parameters are shared across language pairs. We use an effective batch sizes of 500k tokens, and utilize data parallelism to train all models over 64 TPUv3 chips. Sentences are encoded using a shared source-target Sentence Piece Model (Kudo & Richardson, 2018) with 64k tokens, and a `<2xx>` token is prepended to the source sentence to indicate the target language (Johnson et al., 2017). The full training details can be found in Appendix B.

To study real-world multi-task optimization on a massive scale, we use an in-house training corpus[1] (Arivazhagan et al., 2019) generated by crawling and extracting parallel sentences from the web (Uszkoreit et al., 2010), which contains more than 25 billion sentence pairs for 102 languages to and from English. We select 25 languages (50 language pairs pivoted on English), containing over 8 billion sentence pairs, from 10 diverse language families and 4 different levels of data sizes (detailed in Appendix A). We then train two models on two directions separately, namely *Any→En* and *En→Any*. Furthermore, to minimize the confounding factors of inconsistent sentence semantics across language pairs, we create a multi-way aligned evaluation set of 3k sentences for all languages[2]. Then, for each checkpoint at an interval of 1000 training steps, we measure pair-wise cosine similarities of the model's gradients on this dataset between all language pairs. We examine gradient similarities at various granularities, from specific layers to the entire model.

---

[1]We also experiment on publicly available dataset of WMT and obtain similar observations in Appendix C.

[2]In other words, 3k semantically identical sentences are given in 25 languages.

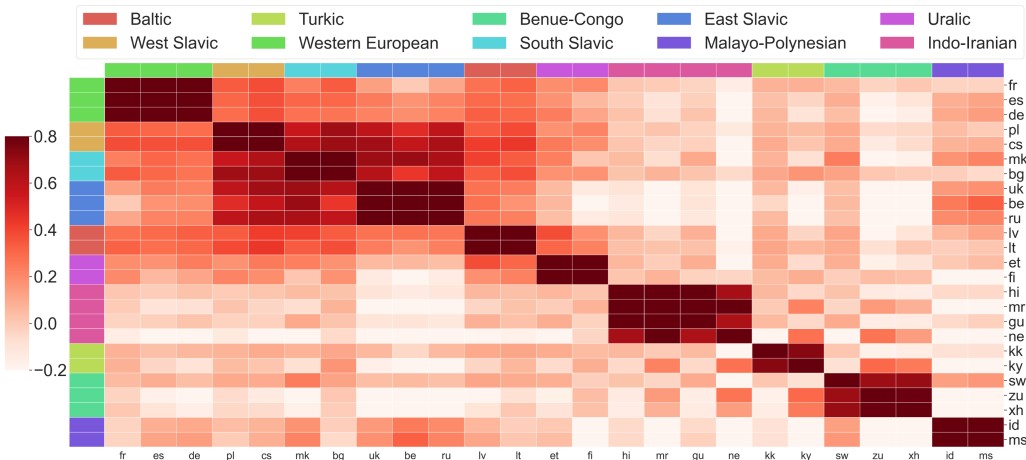

Figure 1: Cosine similarities of encoder gradients between *xx-en* language pairs averaged across all training steps. Darker cell indicates pair-wise gradients are more similar. Best viewed in color.[4]

## 2.2 OBSERVATIONS

We make the following three main observations. Our findings are consistent across different model architectures and settings (see Appendix C and D for more results and additional discussions).

1. **Gradient similarities reflect language proximities.** We first examine if close tasks enjoy similar loss geometries and vice versa. Here, we use language proximity (defined according to their memberships in a linguistic language family) to control task similarity, and utilize gradient similarity to measure loss geometry. We choose typological similarity because it is informative and popular, and we leave the exploration of other language similarity measurements for future work. In Figure 1, we use a symmetric heatmap to visualize pair-wise gradient similarities, averaged across all checkpoints at different training steps. Specifically, we observe strong clustering by membership closeness in the linguistic family, along the diagonal of the gradient similarity matrix. In addition, all European languages form a large cluster in the upper-left corner, with an even smaller fine-grained cluster of Slavic languages inside. Furthermore, we also observe similarities for Western European languages gradually decrease in West Slavic→South Slavic→East Slavic, illustrating the gradual continuum of language proximity.

2. **Gradient similarities correlate positively with model quality.** As gradient similarities correlate well with task proximities, it is natural to ask whether higher gradient similarities lead to better multi-task performance. In Figure 2(a), we train a joint model of all language pairs in both *En→Any* and *Any→En* directions, and compare gradient similarities between these two. While prior work has shown that *En→Any* is harder and less amenable for positive transfer (Arivazhagan et al., 2019), we find that gradients of tasks in *En→Any* are indeed less similar than those in *Any→En*. On the other hand, while larger batch sizes often improve model quality, we observe that models trained with smaller batches have less similar loss geometries (Appendix D). These all indicate that gradient interference poses great challenge to the learning procedure.

   To further verify this, we pair En→Fr with different language pairs (e.g. En→Es or En→Hi), and train a set of models with exactly two language pairs[5]. We then evaluate their performance on the En→Fr test set, and compare their BLEU scores versus gradient similarities between paired two tasks. As shown in Figure 2(b), gradient similarities correlate positively with model performance, again demonstrating that dissimilar gradients introduce interference and undermine model quality.

3. **Gradient similarities evolve across layers and training steps.** While the previous discussion focuses on the gradient similarity of the whole model averaged over all checkpoints, we now

---

[4]Western European includes Romance and Germanic.

[5]To remove confounding factors, we fix the same sampling strategy for all these models.

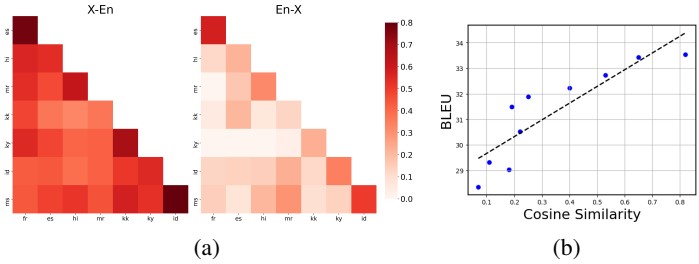

(a)                                     (b)

Figure 2: Comparing gradient similarity versus model performance. **(a):** Similarity of model gradients between *xx-en* (left) and *en-xx* (right) language pairs in a single *Any→Any* model. **(b):** BLEU scores on *en-fr* of a set of trilingual models versus their gradient similarities. Each model is trained on *en-fr* and another *en-xx* language pair.

study it across different layers and training steps. Figure 4(c) shows the evolution of the gradient similarities throughout the training. Interestingly, we observe diverse patterns for different gradient subsets. For instance, gradients between En→Fr and En→Hi gradually become less similar (from positive to negative) in layer 1 of the decoder but more similar (from negative to positive) in the encoder of the same layer. On the other hand, gradient similarities between En→Fr and En→Es are always higher than those between En→Fr and En→Hi in the same layer, consistent with prior observation that gradients reflect language similarities.

In addition, we evaluate the difference between gradient similarities in the multilingual encoder and decoder in Figure 4(a). We find that the gradients are more similar in the decoder (positive values) for the *Any→En* direction but less similar (negative values) for the *En→Any* direction. This is in line with our intuition that gradients should be more consistent when the decoder only needs to handle one single language. Moreover, we visualize how gradient similarities evolve across layers in Figure 4(b). We notice that similarity between gradients increase/decrease as we move up from bottom to top layers for the *Any→En/En→Any* direction, and hypothesize that this is due to the difference in label space (English-only tokens versus tokens from many languages). These results demonstrate that the dynamics of gradients evolve over model layers and training time.

Our analysis highlights the important role of loss geometries in multilingual models. With these points in mind, we next turn to the problem of how to improve multi-task optimization in multilingual models in a systematic way.

## 3 PROPOSED METHOD

Following our observations that inter-task loss geometries correlate well with language similarities and model quality, a natural question to ask next is how we can take advantage of such gradient dynamics and design optimization procedures superior to the standard monolithic practice. Since we train large-scale models on real-world dataset consisting of billions of words, of which tasks are highly unbalanced and exhibit complex interactions, we propose an effective approach that not only exploits inter-task structures but also is applicable to unbalanced tasks and noisy data. To motivate our method, we first review a state-of-the-art multi-task learning method and show how the observation in Section 2 helps us to identify its limitation.

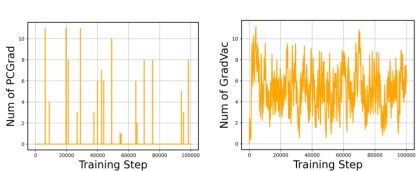

Figure 3: Counts of active PCGrad (left) and GradVac (right) during the training process.

### 3.1 GRADIENT SURGERY

An existing line of work (Chen et al., 2018b; Sener & Koltun, 2018; Yu et al., 2020) has successfully utilized gradient-based techniques to improve multi-task models. Notably, Yu et al. (2020)

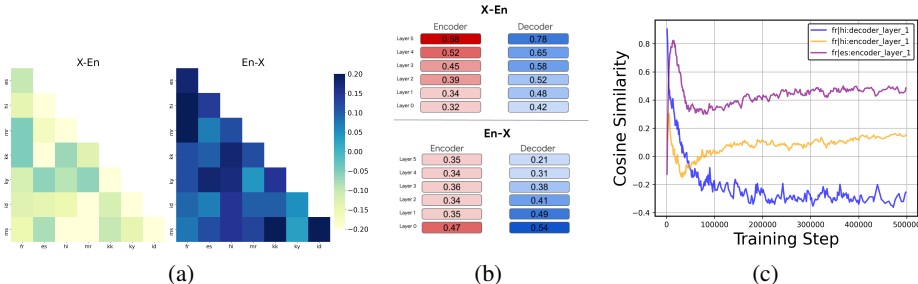

(a)           (b)           (c)

Figure 4: Evaluating gradient similarity across model architecture and training steps. **(a):** Difference between gradient similarities in the encoder and decoder. Positive value (darker) indicates the encoder has more similar gradient similarities. **(b):** Gradient similarities across layers. **(c):** Gradient similarities of different components and tasks across training steps.

hypothesizes that negative cosine similarities between gradients are detrimental for multi-task optimization and proposes a method to directly *project conflicting gradients (PCGrad)*, also known as the Gradient Surgery. As illustrated in the left side of Figure 5(a), the idea is to first detect gradient conflicts and then perform a "surgery" to deconflict them if needed. Specifically, for gradients $\mathbf{g}_i$ and $\mathbf{g}_j$ of the $i$-th and $j$-th task respectively at a specific training step, PCGrad (1) computes their cosine similarity to determine if they are conflicting, and (2) if the value is negative, projects $\mathbf{g}_i$ onto the normal plane of $\mathbf{g}_j$ as:

$$\mathbf{g}'_i = \mathbf{g}_i - \frac{\mathbf{g}_i \cdot \mathbf{g}_j}{\|\mathbf{g}_j\|^2} \mathbf{g}_j. \tag{1}$$

The altered gradient $\mathbf{g}'_i$ replaces the original $\mathbf{g}_i$ and this whole process is repeated across all tasks in a random order. For more details and theoretical analysis, we refer readers to the original work.

Now, we can also interpret PCGrad from a different perspective: notice that the gradient cosine similarity will always be zero after the projection, effectively setting a target lower bound. In other words, PCGrad aims to align gradients to match a certain gradient similarity level, and implicitly makes the assumption that *any two tasks must have the same gradient similarity objective of zero*. However, as we shown in Section 2, different language proximities would result in diverse gradient similarities. In fact, many language pairs in our model share positive cosine similarities such that the pre-condition for PCGrad would never be satisfied. This is shown in the left of Figure 5(b), where PCGrad is not effective for positive gradient similarities and thus it is very sparse during training in the left of Figure 3. Motivated by this limitation, we next present our proposed method.

### 3.2 Gradient Vaccine

The limitation of PCGrad comes from the unnecessary assumption that all tasks must enjoy similar gradient interactions, ignoring complex inter-task relationships. To relax this assumption, a natural idea is to set adaptive gradient similarity objectives in some proper manner. An example is shown in the right of Figure 5(b), where two tasks have a positive gradient similarity of $\cos(\theta) = \phi_{ij}$. While PCGrad ignores such non-negative case, the current value of $\phi_{ij}$ may still be detrimentally low for more similar tasks such as French versus Spanish. Thus, suppose we have some similarity goal of $\cos(\theta') = \phi_{ij}^T > \phi_{ij}$ (e.g. the "normal" cosine similarity between these two tasks), we alter both the magnitude and direction of $\mathbf{g}_i$ such that the resulting gradients match such gradient similarity objective. In particular, we replace $g_i$ with a vector that satisfies such condition in the vector space spanned by $\mathbf{g}_i$ and $\mathbf{g}_j$, i.e. $a_1 \cdot \mathbf{g}_i + a_2 \cdot \mathbf{g}_j$. Since there are infinite numbers of valid combinations of $a_1$ and $a_2$, for simplicity, we fix $a_1 = 1$ and by applying Law of Sines in the plane of $\mathbf{g}_i$ and $\mathbf{g}_j$, we solve for the value of $a_2$ and derive the new gradient for the $i$-th task as [6]:

$$\mathbf{g}'_i = \mathbf{g}_i + \frac{\|\mathbf{g}_i\|(\phi_{ij}^T \sqrt{1 - \phi_{ij}^2} - \phi_{ij}\sqrt{1 - (\phi_{ij}^T)^2})}{\|\mathbf{g}_j\|\sqrt{1 - (\phi_{ij}^T)^2}} \cdot \mathbf{g}_j. \tag{2}$$

---

[6]See Appendix E for derivation detail, implementation in practice and theoretical analysis.

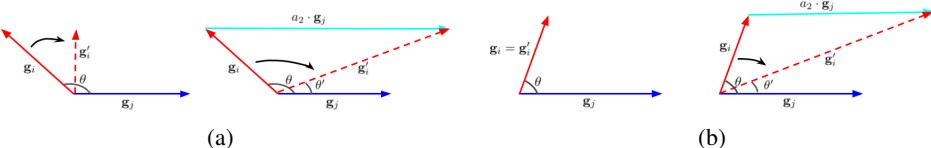

Figure 5: Comparing PCGrad (left) with GradVac (right) in two cases. **(a):** For negative similarity, both methods are effective but GradVac can utilize adaptive objectives between different tasks. **(b):** For positive similarity, only GradVac is active while PCGrad stays "idle".

This formulation allows us to use arbitrary gradient similarity objective $\phi_{ij}^T$ in $[-1, 1]$. The remaining question is how to set such objective properly. In the above analysis, we have seen that gradient interactions change drastically across tasks, layers and training steps. To incorporate these three factors, we exploit an exponential moving average (EMA) variable for tasks $i, j$ and parameter group $k$ (e.g. the $k$-th layer) as:

$$\hat{\phi}_{ijk}^{(t)} = (1 - \beta)\hat{\phi}_{ijk}^{(t-1)} + \beta\phi_{ijk}^{(t)}, \tag{3}$$

where $\phi_{ijk}^{(t)}$ is the computed gradient similarity at training step $t$, $\beta$ is a hyper-parameter, and $\hat{\phi}_{ijk}^{(0)} = 0$. The full method is outlined in Algorithm 1 (Appendix E). Notice that gradient surgery is a special case of our proposed method such that $\phi_{ij}^T = 0$. As shown in the right of Figure 5(a) and 5(b), our method alters gradients more *preemptively* under both positive and negative cases, taking more proactive measurements in updating the gradients (Figure 3). We therefore refer to it as **Gradient Vaccine (GradVac)**. Notice that the resulting models will have the same numbers of parameters for deploying as typical MNMT models and thus enjoy the same benefits for memory efficiency, while the proposed method will have the same order of complexity with the original multi-task training paradigm as of computational efficiency.

## 4 EXPERIMENTS

We compare multi-task optimization methods with the monolithic approach in multilingual settings, and examine the effectiveness of our proposed method on multilingual NMT and multilingual language models.

### 4.1 GENERAL SETUP

We choose three popular scalable gradient-based multi-task optimization methods as our baselnes: **GradNorm** (Chen et al., 2018b), **MGDA** (Sener & Koltun, 2018), and **PCGrad** (Yu et al., 2020). For fair comparison, language-specifc gradients are computed for samples in each batch. The sampling temperature is also fixed at T=5 unless otherwise stated. For the baselines, we mainly follow the default settings and training procedures for hype-parameter selection as explained in their respective papers. For our method, to study how sensitive GradVac is to the distribution of tasks, we additionally examine a variant that allows us to control which languages are considered for GradVac. Specifically, we search the following hyper-parameters on small-scale WMT dataset and transfer to our large-scale dataset: tasks considered for GradVac {HRL_only, LRL_only, all_task}, parameter granularity {whole_model, enc_dec, all_layer, all_matrix}, EMA decay rate $\beta$ {1e-1, 1e-2, 1e-3}. We find {LRL_only, all_layer, 1e-2} to work generally well and use these in the following experiments (see Appendix F for more details and results).

### 4.2 RESULTS AND ANALYSIS

**WMT Machine Translation.** We first conduct comprehensive analysis of our method and other baselines on a small-scale WMT task. We consider two high-resource languages (WMT14 en-fr, WMT19 en-cs) and two low-resource languages (WMT14 en-hi, WMT18 en-tr), and train two models for both to and from English. Results are shown in Table 1.

| | En→Any | | | | | Any→En | | | | |
|---|---|---|---|---|---|---|---|---|---|---|
| | en-fr | en-cs | en-hi | en-tr | avg | fr-en | cs-en | hi-en | tr-en | avg |
| Monolithic Training | | | | | | | | | | |
| (1) Bilingual Model | 41.80 | 24.76 | 5.77 | 9.77 | 20.53 | 36.38 | 29.17 | 8.68 | 13.87 | 22.03 |
| (2) Multilingual Model | 37.24 | 20.22 | 13.69 | 18.77 | 22.48 | 34.29 | 27.66 | 18.48 | 22.01 | 25.61 |
| Multi-task Training | | | | | | | | | | |
| (3) GradNorm (Chen et al., 2018b) | 37.02 | 18.78 | 11.57 | 15.44 | 20.70 | 34.58 | 27.85 | 18.03 | 22.37 | 25.71 |
| (4) MGDA (Sener & Koltun, 2018) | 38.22 | 17.54 | 12.02 | 13.69 | 20.37 | 35.05 | 26.87 | 18.28 | 22.41 | 25.65 |
| (5) PCGrad (Yu et al., 2020) | 37.72 | 20.88 | 13.77 | 18.23 | 22.65 | 34.37 | 27.82 | 18.78 | 22.20 | 25.79 |
| (6) PCGrad w. all_layer | 38.01 | 21.04 | 13.95 | 18.46 | 22.87 | 34.57 | 27.84 | 18.84 | 22.48 | 25.93 |
| Our Approach | | | | | | | | | | |
| (7) GradVac w. fixed_obj | 38.41 | 21.12 | 13.75 | 18.68 | 22.99 | 34.55 | 27.97 | 18.72 | 22.14 | 25.85 |
| (8) GradVac w. whole_model | 38.76 | 21.32 | 14.22 | 18.89 | 23.30 | 34.84 | 28.01 | 18.85 | 22.24 | 25.99 |
| (9) GradVac w. all_layer | **39.27*** | **21.67*** | 14.88* | **19.73*** | 23.89 | **35.28*** | **28.42*** | 19.07* | 22.58* | 26.34 |

Table 1: BLEU scores on the WMT dataset. The best result for multilingual model is **bolded** while underline signifies the overall best, and * means the gains over baseline multilingual models are statistically significant with p < 0.05.

First, we observe that while the naive multilingual baseline outperforms bilingual models on low-resource languages, it performs worse on high-resource languages due to negative interference (Wang et al., 2020b) and constrained capacity (Arivazhagan et al., 2019). Existing baselines fail to address this problem properly, as they obtain marginal or even no improvement (row 3, 4 and 5). In particular, we look closer at the optimization process for methods that utilize gradient signals to reweight tasks, i.e. GradNorm and MGDA, and find that their computed weights are less meaningful and noisy. For example, MGDA assigns larger weight for en-fr in the en-xx model, that results in worse performance on other languages. This is mainly because these methods are designed under the assumption that all tasks have balanced data. Our results show that simply reweighting task weights without considering the loss geometry has limited efficacy.

By contrast, our method significantly outperforms all baselines. Compared to the naive joint training approach, the proposed method improves over not only the average BLEU score but also the individual performance on all tasks. We notice that the performance gain on *En→Any* is larger compared to *Any→En*. This is in line with our prior observation that gradients are less similar and more conflicting in *En→Any* directions.

We next conduct extensive ablation studies for deeper analysis: (1) GradVac applied to all layers vs. whole model (row 8 vs. 9): the all_layer variant outperforms whole_model, showing that setting fine-grained parameter objectives is important. (2) Constant objective vs. EMA (row 7 vs. 9): we also examine a variant of GradVac optimized using a constant gradient objective for all tasks (e.g. $\phi_{ij}^T = 0.5, \forall i, j$) and observe performance drop compared to using EMA variables. This highlights the importance of setting task-aware objectives through task relatedness. (3) GradVac vs. PCGrad (row 8-9 vs. 5-6): the two GradVac variants outperform their PCGrad counterparts, validating the effectiveness of setting preemptive gradient similarity objectives.

**Massively Multilingual Machine Translation.** We then scale up our experiments and transfer the best setting found on WMT to the same massive dataset used in Section 2. We visualize model performance in Figure 6 and average BLEU scores are shown in Table 2. We additionally compare with models trained with uniform language pairs sampling strategy (T=1) and find that our method outperforms both multilingual models. Most notably, while uniform sampling favor high-resource language pairs more than low-resource ones, GradVac is able to improve both consistently across all tasks. We observe larger performance gain on

| Any→En | High | Med | Low | All |
|---|---|---|---|---|
| T=1 | 28.56 | 28.51 | 19.57 | 24.95 |
| T=5 | 28.16 | 28.42 | 24.32 | 26.71 |
| GradVac | **28.99** | **28.94** | **24.58** | **27.21** |

| En→Any | High | Med | Low | All |
|---|---|---|---|---|
| T=1 | 22.62 | 21.53 | 12.41 | 18.18 |
| T=5 | 22.04 | 21.43 | 13.07 | 18.25 |
| GradVac | **24.20** | **21.83** | **13.30** | **19.08** |

Table 2: Average BLEU scores of 25 language pairs on our massively multilingual dataset.

high-resource languages, illustrating that addressing gradient conflicts can mitigate negative interference on these head language pairs. On the other hand, our model still perform worse on resourceful languages compared to bilingual baselines, most likely limited by model capacity.

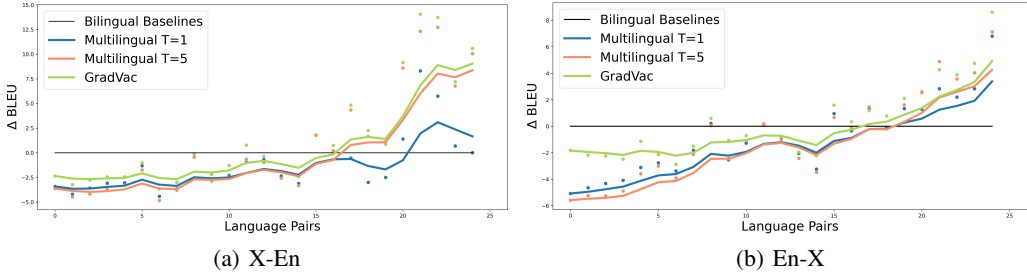

| | (a) X-En | (b) En-X |
|---|---|---|

Figure 6: Comparing multilingual models with bilingual baselines on our dataset. Language pairs are listed in the order of training data sizes (high-resource languages on the left).

| | de | en | es | hi | jv | kk | mr | my | sw | te | tl | yo | avg |
|---|---|---|---|---|---|---|---|---|---|---|---|---|---|
| mBERT | 83.2 | 77.9 | 87.5 | 82.2 | 77.6 | 87.6 | 82.0 | **75.8** | 87.7 | 78.9 | 83.8 | 90.7 | 82.9 |
| + GradNorm | 83.5 | 77.4 | 87.2 | 82.7 | 78.4 | **87.9** | 81.2 | 73.4 | 85.2 | 78.7 | 83.6 | 91.5 | 82.6 |
| + MGDA | 82.1 | 74.2 | 85.6 | 81.5 | 77.8 | 87.8 | 81.9 | 74.3 | 86.5 | 78.2 | 87.5 | 91.7 | 82.4 |
| + PCGrad | 83.7 | 78.6 | 88.2 | 81.8 | 79.6 | 87.6 | 81.8 | 74.2 | 85.9 | 78.5 | 85.6 | 92.2 | 83.1 |
| + GradVac | **83.9** | **79.4** | **88.2** | **81.8** | **80.5** | 87.4 | **82.1** | 73.9 | **87.8** | **79.3** | 87.8 | **93.0** | **83.8** |

Table 3: F1 on the NER tasks of the XTREME benchmark.

**XTREME Benchmark.** We additionally apply our method to multilingual language models and evaluate on the XTREME benchmark (Hu et al., 2020). We choose tasks where training data are available for all languages, and finetune a pretrained multilingual BERT model (mBERT) (Devlin et al., 2018) on these languages jointly (see Appendix G for experiment details and additional results). As shown in Table 3, our method consistently outperforms naive joint finetuning and other multi-task baselines. This demonstrates the practicality of our approach for general multilingual tasks.

## 5 RELATED WORK

Multilingual models train multiple languages jointly (Firat et al., 2016; Devlin et al., 2018; Lample & Conneau, 2019; Conneau et al., 2019; Johnson et al., 2017; Aharoni et al., 2019; Arivazhagan et al., 2019). Follow-up work study the cross-lingual ability of these models and what contributes to it (Pires et al., 2019; Wu & Dredze, 2019; Wu et al., 2019; Artetxe et al., 2019; Kudugunta et al., 2019; Karthikeyan et al., 2020), the limitation of such training paradigm (Arivazhagan et al., 2019; Wang et al., 2020b), and how to further improve it by utilizing post-hoc alignment (Wang et al., 2020c; Cao et al., 2020), data balancing (Jean et al., 2019; Wang et al., 2020a), or calibrated training signal (Mulcaire et al., 2019; Huang et al., 2019a). In contrast to these studies, we directly investigate language interactions across training progress using loss geometry and propose a language-aware method to improve the optimization procedure.

On the other hand, multilingual models can be treated as multi-task learning methods (Ruder, 2017; Zamir et al., 2018). Prior work have studied the optimization challenges of multi-task training (Hessel et al., 2019; Schaul et al., 2019), while others suggest to improve training quality through learning task relatedness (Zhang & Yeung, 2012), routing task-specifc paths (Rusu et al., 2016; Rosenbaum et al., 2019), altering gradients directly (Kendall et al., 2018; Chen et al., 2018b; Du et al., 2018; Yu et al., 2020), or searching pareto solutions (Sener & Koltun, 2018; Lin et al., 2019). However, while these methods are often evaluated on balanced task distributions, multilingual datasets are often unbalanced and noisy. As prior work have shown training with unbalanced tasks can be prone to negative interference (Ge et al., 2014; Wang & Carbonell, 2018), we study how to mitigate it in large models trained with highly unbalanced and massive-scale dataset.

## 6    CONCLUSION

In this paper, we systematically study loss geometry through the lens of gradient similarity for multilingual modeling, and propose a novel approach named GradVac for improvement based on our findings. Leveraging the linguistic proximity structure of multilingual tasks, we validate the assumption that more similar loss geometries improve multi-task optimization while gradient conflicts can hurt model performance, and demonstrate the effectiveness of more geometrically consistent updates aligned with task closeness. We analyze the behavior of the proposed approach on massive multilingual tasks with superior performance, and we believe that our approach is generic and applicable beyond multilingual settings.

ACKNOWLEDGMENTS

We want to thank Hieu Pham for tireless help to the authors on different stages of this project. We also would like to thank Zihang Dai, Xinyi Wang, Zhiyu Wang, Jiateng Xie, Yiheng Zhou, Ruochen Xu, Adams Wei Yu, Biao Zhang, Isaac Caswell, Sneha Kudugunta, Zhe Zhao, Christopher Fifty, Xavier Garcia, Ye Zhang, Macduff Hughes, Yonghui Wu, Samy Bengio and the Google Brain team for insightful discussions and support to the work. This material is based upon work supported in part by the National Science Foundation under Grants No. IIS2007960 and IIS2040926, and by the Google faculty research award.

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

## A  DATA STATISTICS

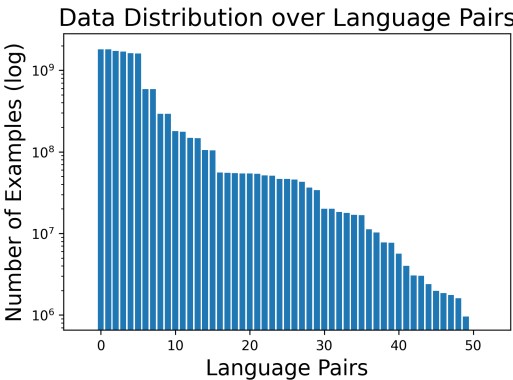

Figure 7: Per language pair data distribution of the dataset used to train our multilingual model. The yaxis depicts the number of training examples available per language pair on a logarithmic scale.

We select 25 languages (50 language pairs) from our dataset to be used in our multilingual models for more careful studies on gradient trajectory. For such purpose, we pick languages that belong to different language families (typologically diverse) and with various levels of training data sizes. Specifically, we consider the following languages and their details are listed in 4: French (fr), Spanish (es), German (de), Polish (pl), Czech (cs), Macedonian (mk), Bulgarian (bg), Ukrainian (uk), Belarusian (be), Russian (ru), Latvian (lv), Lithuanian (lt), Estonian (et), Finnish (fi), Hindi (hi), Marathi (mr), Gujarati (gu), Nepali (ne), Kazakh (kk), Kyrgyz (ky), Swahili (sw), Zulu (zu), Xhosa (xh), Indonesian (id), Malay (ms).

Our corpus has languages belonging to a wide variety of scripts and linguistic families. The selected 25 languages belong to 10 different language families (e.g. Turkic versus Uralic) or branches within language family (e.g. East Slavic versus West Slavic), as indicated in Figure 1 and Table 4. Families are groups of languages believed to share a common ancestor, and therefore tend to have similar vocabulary and grammatical constructs. We therefore utilize membership of language family to define language proximity.

In addition, our language pairs have different levels of training data, ranging from $10^5$ to $10^9$ sentence pairs. This is shown in Figure 7. We therefore have four levels of data sizes (number of languages in parenthesis): High (7), Medium (8), Low (5), and Extremely Low (5). In particular, we consider tasks with more than $10^8$ to be high-resource, $10^7 - 10^8$ to be medium-resource, and rest to be low-resource (with those below 5 million sentence pairs to be extremely low-resource). Therefore, our dataset is both heavily unbalanced and noisy, as it is crawled from the web, and thus introduces optimization challenges from a multi-task training perspective. These characteristics of our dataset make the problem that we study as realistic as possible.

## B  TRAINING DETAILS

For both bilingual and multilingual NMT models, we utilize the encoder-decoder Transformer (Vaswani et al., 2017) architecture. Following prior work, we share all parameters across all language pairs, including word embedding and output softmax layer.

To train each model, we use a single Adam optimizer (Kingma & Ba, 2014) with default decay hyper-parameters. We warm up linearly for 30K steps to a learning rate of 1e-3, which is then decayed with the inverse square root of the number of training steps after warm-up. At each training step, we sample from all language pairs according to a temperature based sampling strategy as in prior work (Lample & Conneau, 2019; Arivazhagan et al., 2019). That is, at each training step, we sample each sentence from all language pairs to train proportionally to $P_i = \left(\frac{L_i}{\sum_j L_j}\right)^{\frac{1}{T}}$, where $L_i$ is the size of the training corpus for language pair i and T is the temperature. We set T=5 for most of our experiments.

| Language | Id | Language Family | Data Size | Language | Id | Language Family | Data Size |
|---|---|---|---|---|---|---|---|
| French | fr | Western European | High | Finnish | fi | Uralic | High |
| Spanish | es | Western European | High | Hindi | hi | Indo-Iranian | Medium |
| German | de | Western European | High | Marathi | mr | Indo-Iranian | Ex-Low |
| Polish | pl | West Slavic | High | Gujarati | gu | Indo-Iranian | Low |
| Czech | cs | West Slavic | High | Nepali | ne | Indo-Iranian | Ex-Low |
| Macedonian | mk | South Slavic | Low | Kazakh | kk | Turkic | Low |
| Bulgarian | bg | South Slavic | Medium | Kyrgyz | ky | Turkic | Ex-Low |
| Ukrainian | uk | East Slavic | Medium | Swahili | sw | Benue-Congo | Low |
| Belarusian | be | East Slavic | Low | Zulu | zu | Benue-Congo | Ex-Low |
| Russian | ru | East Slavic | High | Xhosa | xh | Benue-Congo | Ex-Low |
| Latvian | lv | Baltic | Medium | Indonesian | id | Malayo-Polynesian | High |
| Lithuanian | lt | Baltic | Medium | Malay | ms | Malayo-Polynesian | Medium |
| Estonian | et | Uralic | Medium | | | | |

Table 4: Details of all languages considered in our dataset. Notice that since German (Germanic) is particularly similar to French and Spanish (Romance), we consider a larger language branch for them named "Western European". "Ex-Low" indicates extremely low-resource languages in our dataset. We use BCP-47 language codes as labels (Phillips & Davis, 2006).

| Language | Id | Language Family | Data Size | Validation Set |
|---|---|---|---|---|
| French | fr | Romance | 41M | newstest2013 |
| Spanish | es | Romance | 15M | newstest2012 |
| Russian | ru | Slavic | 38M | newstest2018 |
| Czech | cs | Slavic | 37M | newstest2018 |
| Latvian | lv | Baltic | 6M | newstest2017 |
| Lithuanian | lt | Baltic | 6M | newstest2019 |
| Estonian | et | Uralic | 2M | newstest2018 |
| Finnish | fi | Uralic | 6M | newstest2018 |

Table 5: Details of all languages selected from WMT for gradient analysis.

## C  ADDITIONAL RESULTS ON WMT

### C.1  DATA

We experiment with WMT datasets that are publicly available. Compared to our dataset, they only contain a relatively small subsets of languages. Therefore, we select 8 languages (16 language pairs) of 4 language families to conduct the same loss geometries analysis in Section 2. These languages are detailed in Table x5: French (fr), Spanish (es), Russian (ru), Czech (cs), Latvian (lv), Lithuanian (lt), Estonian (et), Finnish (fi). We collect all available training data from WMT 13 to WMT 19, and then perform a deduplication process to remove duplicated sentence pairs. We then use the validation sets to compute gradient similarities. Notice that unlike our dataset, WMT validation sets are not multi-aligned. Therefore, the semantic structures of these sentences may introduce an extra degree of noise.

### C.2  VISUALIZATION

As in Section 2, we compute gradients on the validation sets on all checkpoints and averaged across all checkpoints to visualize our results. We use similar setups to our previous analysis, including model architectures, vocabulary sizes, and other training details. The main results are shown in Figure 8. Similar to our findings in Section 2, gradient similarities cluster according to language proximities, with languages from the same language family sharing the most similar gradients on the diagonal. Besides, gradients in the English to Any directions are less similar compared to the other direction, consistent with our above findings. Overall, despite the scale being much smaller in terms of number of languages and sizes of training data, findings are mostly consistent.

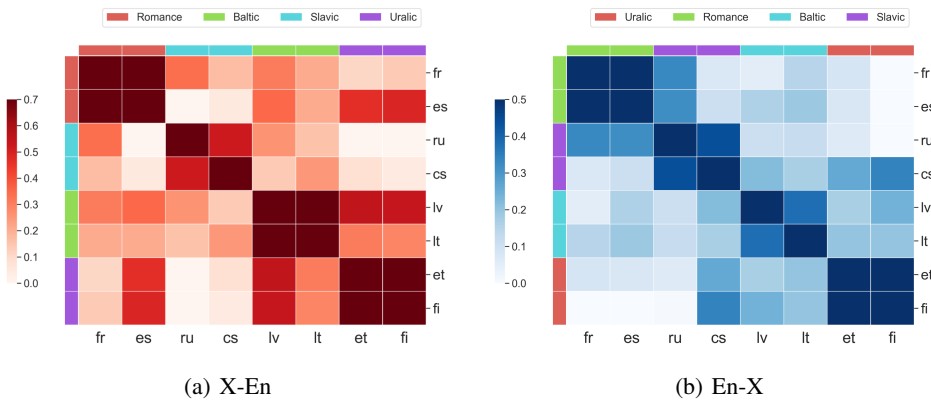

(a) X-En  (b) En-X

Figure 8: Cosine similarities on WMT dataset averaged across all training steps.

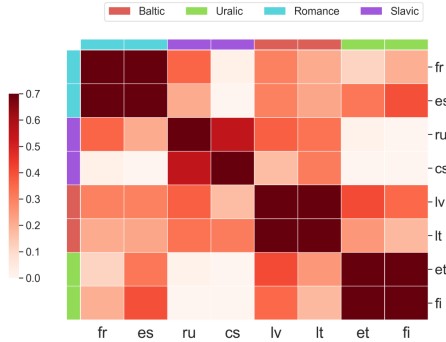

Figure 9: Cosine similarities (on Transformer-Base models) of *xx-en* language pairs on WMT dataset averaged across all training steps.

## C.3 VISUALIZATION ON SMALLER MODELS

Prior work has shown languages fighting for capacity in multilingual models (Arivazhagan et al., 2019; Wang et al., 2020b). Therefore, we are also interested to study the effect of model sizes on gradient trajectory. Since our larger dataset contains 25 language pairs in a Transformer-Large model, we additionally train a Transformer-Base model using the 8 language pairs of WMT. We visualize it in Figure 9 and find that our observed patterns are more evident in smaller models. This finding is consistent across other experiments we ran and indicates that languages compete for capacity with small model sizes thereby causing more gradient interference. It also shows that our analysis in this work is generic across different model settings.

## D ADDITIONAL RESULTS ON OUR DATASET

In Figure 1 we show visualization on models trained using *Any→En* language pairs. Here, we also examine models trained in the other direction, *En→Any*. As shown in Figure 10, we have similar observations made in Section 2 such that gradient similarities cluster strongly by language proximities. However, the *en-xx* model has smaller scales in cosine similarities and more negative values. For example, Nepali shares mostly conflicting gradients with other languages, except for those belonging to the same language family. This is in line with our above discussion that gradient interference may be a source of optimization challenge, such that the *en-xx* model is harder to train than the *xx-en* model.

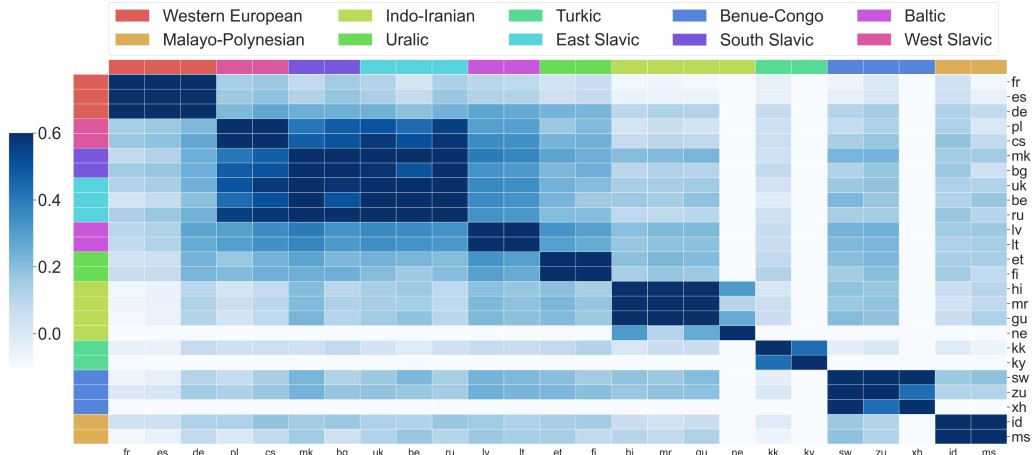

Figure 10: Cosine similarities of decoder gradients between *en-xx* language pairs averaged across all training steps. Darker cell indicates pair-wise gradients are more similar.

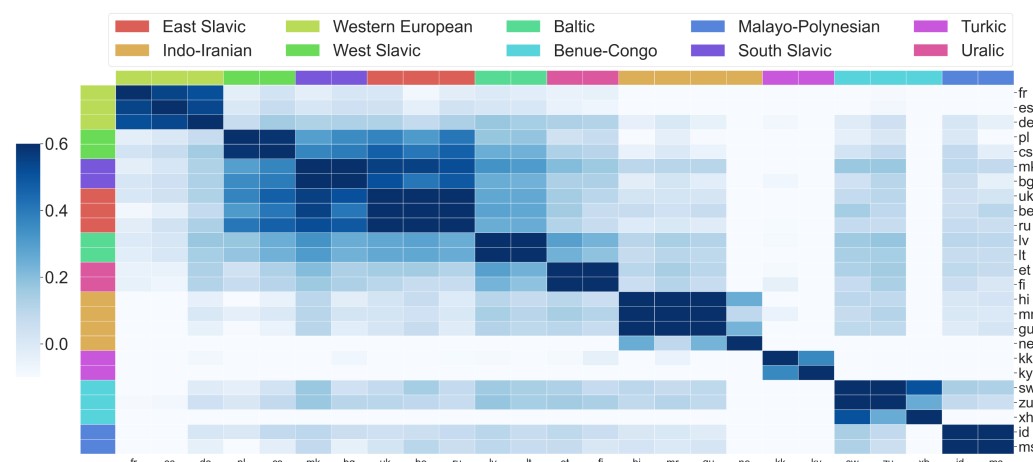

Figure 11: Cosine similarities of decoder gradients between *en-xx* language pairs averaged across all training steps. Darker cell indicates pair-wise gradients are more similar. Model trained with smaller batch sizes.

Moreover, while our previous models are trained using a large batch size for better performance (as observed in prior work (Arivazhagan et al., 2019)), we also evaluate gradients in a model trained with smaller batches (125k tokens) in Figure 11. Compared to model trained with larger batch sizes, this model enjoy similar patterns but with smaller gradient cosine similarity values, indicating that gradients are less similar. This presents an additional potential explanation of why larger batch sizes can be more effective for training large models: they may better reflect the correct loss geometries such that gradients are less conflicting in nature. For our case, this means larger batches better reflect language proximities hence gradients of better quality.

Finally, these results also reveal that gradient similarities are mostly dependent on task relatedness, as even sentence pairs with identical semantic meanings can have negative cosine similarities due to language differences.

---

**Algorithm 1** GradVac Update Rule

---

1: **Require:** EMA decay $\beta$, Model Components $\mathcal{M} = \{\boldsymbol{\theta}_k\}$, Tasks for GradVac $\mathcal{G} = \{\mathcal{T}_i\}$
2: Initialize model parameters
3: Initialize EMA variables $\hat{\phi}_{ijk}^{(0)} = 0, \forall i, j, k$
4: Initialize time step $t = 0$
5: **while** not converged **do**
6:     Sample minibatch of tasks $\mathcal{B} = \{\mathcal{T}_i\}$
7:     **for** $\boldsymbol{\theta}_k \in \mathcal{M}$ **do**
8:         Compute gradients $\mathbf{g}_{ik} \leftarrow \nabla_{\boldsymbol{\theta}_k} \mathcal{L}_{\mathcal{T}_i}, \forall \mathcal{T}_i \in \mathcal{B}$
9:         Set $\mathbf{g}'_{ik} \leftarrow \mathbf{g}_{ik}$
10:         **for** $\mathcal{T}_i \in \mathcal{G} \cap \mathcal{B}$ **do**
11:           **for** $\mathcal{T}_j \in \mathcal{B} \setminus \mathcal{T}_i$ in random order **do**
12:             Compute $\phi_{ijk}^{(t)} \leftarrow \frac{\mathbf{g}'_{ik} \cdot \mathbf{g}_{jk}}{\|\mathbf{g}'_{ik}\| \|\mathbf{g}_{jk}\|}$
13:             **if** $\phi_{ijk}^{(t)} < \hat{\phi}_{ijk}^{(t)}$ **then**
14:                Set $\mathbf{g}'_{ik} = \mathbf{g}'_{ik} + \frac{\|\mathbf{g}'_{ik}\|(\hat{\phi}_{ijk}^{(t)} \sqrt{1-(\phi_{ijk}^{(t)})^2} - \phi_{ijk}^{(t)} \sqrt{1-(\hat{\phi}_{ijk}^{(t)})^2})}{\|\mathbf{g}_{jk}\| \sqrt{1-(\hat{\phi}_{ijk}^{(t)})^2}} \cdot \mathbf{g}_{jk}$
15:             **end if**
16:             Update $\hat{\phi}_{ijk}^{(t+1)} = (1 - \beta)\hat{\phi}_{ijk}^{(t)} + \beta\phi_{ijk}^{(t)}$
17:           **end for**
18:         **end for**
19:         Update $\boldsymbol{\theta}_k$ with gradient $\sum \mathbf{g}'_{ik}$
20:     **end for**
21:     Update $t \leftarrow t + 1$
22: **end while**

---

# E  PROPOSED METHOD DETAILS

In this part, we provide details of our proposed method, Gradient Vaccine (GradVac). We first show how to derive our formulation in Eq. 2, followed by how we instantiate in practice. And last, we also study its theoretical property.

## E.1  METHOD DERIVATION

As stated in Section 3, the goal of our proposed method is to align gradients between tasks to match a pre-set target gradient cosine similarity. An example is shown in Figure 12, where we have two tasks $i, j$ and their corresponding gradients $\mathbf{g}_i$ and $\mathbf{g}_j$ have a cosine similarity of $\phi_{ij}$, i.e. $\cos(\theta) = \phi_{ij} = \frac{\mathbf{g}_i \cdot \mathbf{g}_j}{\|\mathbf{g}_i\|\|\mathbf{g}_j\|}$. Then, we want to alter their gradients, such that the resulting new gradients have gradient similarity of some pre-set value $\phi_{ij}^T$. To do so, we replace $\mathbf{g}_i$ with a new vector in the vector space spanned by $\mathbf{g}_i$ and $\mathbf{g}_j$, $\mathbf{g}_i' = a_1 \cdot \mathbf{g}_i + a_2 \cdot \mathbf{g}_j$. Without loss of generality, we set $a_1 = 1$ and solve for $a_2$, i.e. find the $a_2$ such that $\cos(\gamma) = \frac{\mathbf{g}_i' \cdot \mathbf{g}_j}{\|\mathbf{g}_i'\|\|\mathbf{g}_j\|} = \phi_{ij}^T$. By using Laws of Sines, we must have that:

$$\frac{\|\mathbf{g}_i\|}{\sin(\gamma)} = \frac{a_2\|\mathbf{g}_j\|}{\sin(\theta - \gamma)}, \tag{4}$$

and thus we can further solve for $a_2$ as:

$$\frac{\|\mathbf{g}_i\|}{\sin(\gamma)} = \frac{a_2\|\mathbf{g}_j\|}{\sin(\theta - \gamma)}$$

$$\Rightarrow \frac{\|\mathbf{g}_i\|}{\sin(\gamma)} = \frac{a_2\|\mathbf{g}_j\|}{\sin(\theta)\cos(\gamma) - \cos(\theta)\sin(\gamma)}$$

$$\Rightarrow \frac{\|\mathbf{g}_i\|}{\sqrt{1 - (\phi_{ij}^T)^2}} = \frac{a_2\|\mathbf{g}_j\|}{\phi_{ij}^T\sqrt{1 - \phi_{ij}^2} - \phi_{ij}\sqrt{1 - (\phi_{ij}^T)^2}}$$

$$\Rightarrow a_2 = \frac{\|\mathbf{g}_i\|(\phi_{ij}^T\sqrt{1 - \phi_{ij}^2} - \phi_{ij}\sqrt{1 - (\phi_{ij}^T)^2})}{\|\mathbf{g}_j\|\sqrt{1 - (\phi_{ij}^T)^2}}$$

We therefore arrive at the update rule in Eq. 2. Our formulation allows us to set arbitrary target values for any two gradients, and thus we can better leverage task relatedness by setting individual gradient similarity objective for each task pair. Notice that we can rescale the gradient such that the altered gradients will have the same norm as before. But in our experiment we find it is sufficient to ignore this step. On the other hand, we note that when $\phi_{ij}^T = 0$, we have that:

$$a_2 = \frac{\|\mathbf{g}_i\|(\phi_{ij}^T\sqrt{1 - \phi_{ij}^2} - \phi_{ij}\sqrt{1 - (\phi_{ij}^T)^2})}{\|\mathbf{g}_j\|\sqrt{1 - (\phi_{ij}^T)^2}}$$

$$= \frac{\|\mathbf{g}_i\|(0 - \phi_{ij})}{\|\mathbf{g}_j\|}$$

$$= -\frac{\mathbf{g}_i \cdot \mathbf{g}_j}{\|\mathbf{g}_i\|\|\mathbf{g}_j\|} \cdot \frac{\|\mathbf{g}_i\|}{\|\mathbf{g}_j\|}$$

$$= -\frac{\mathbf{g}_i \cdot \mathbf{g}_j}{\|\mathbf{g}_j\|^2}$$

This is exactly the update rule of PCGrad in Eq. 1. Thus, PCGrad is a special case of our proposed method.

## E.2  ALGORITHM IN PRACTICE

Our proposed method is detailed in Algorithm 1. In our experiments on multilingual NMT and multilingual BERT, we utilize a set of exponential moving average (EMA) variables to set proper

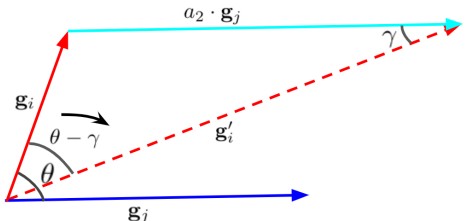

Figure 12: Pictorial description of our method.

pair-wise gradient similarity objectives, as shown in Eq. 3. This is motivated by our observations in Section 2 such that gradients of different languages in a Transformer model evolve across layers and training steps. Therefore, we conduct GradVac on different model components independently. For example, we can do one GradVac on each layer in the model, or just perform a single GradVac on the entire model. In addition, we also introduce an extra degree of freedom by controlling which tasks to perform GradVac. This corresponds to selecting a of tasks $\mathcal{G}$ and only alter gradients for tasks within this set, as shown in line 10 in Algorithm 1. Empirically, we find performing GradVac by layers and on low-resource languages to work generally the best (See Appendix F for detailed discussion).

### E.3 THEORETICAL PROPERTY

Finally, we analyze the theoretical property of our method. Supper we only have two tasks, and their losses are $\mathcal{L}_1$ and $\mathcal{L}_2$, and we denote their gradient cosine similarity at a given step as $\phi_{12}$. When $\phi_{12}$ is negative, our method is largely equivalent to PCGrad and enjoy PCGrad's convergence analysis. Thus, here we consider the other case when $\phi_{12}$ is positive and show that:

**Theorem 1.** *Suppose $\mathcal{L}_1$ and $\mathcal{L}_2$ are convex and differentiable, and that the gradient of $\mathcal{L}$ is Lipschitz continuous with constant $L > 0$. Then, the GradVac update rule with step size[7] $t < \frac{2}{L(1+a^2)}$ and $t < \frac{1}{L}$, where $a = \frac{\sin(\phi_{12}-\phi_{12}^T)}{\sin(\phi_{12}^T)}$ ($\phi_{12} > 0$ and $\phi_{12}^T \geq \phi_{12}$ is some target cosine similarity), will converge to the optimal value $\mathcal{L}(\theta^*)$.*

*Proof.* Let $\mathbf{g}_1 = \nabla \mathcal{L}_1$ and $\mathbf{g}_2 = \nabla \mathcal{L}_2$ be gradients for task 1 and task 2 respectively. Thus we have $\mathbf{g} = \mathbf{g}_1 + \mathbf{g}_2$ as the original gradient and $\mathbf{g}' = \mathbf{g} + a\frac{\|\mathbf{g}_2\|}{\|\mathbf{g}_1\|}\mathbf{g}_1 + a\frac{\|\mathbf{g}_1\|}{\|\mathbf{g}_2\|}\mathbf{g}_2$ as the altered gradient by the GradVac update rule, such that:

$$a = \frac{\sin(\phi_{12})\cos(\phi_{12}^T) - \cos(\phi_{12})\sin(\phi_{12}^T)}{\sin(\phi_{12}^T)} \tag{5}$$

where $\phi_{12}^T$ is some pre-set gradient similarity objective and $\phi_{12}^T \geq \phi_{12}$ (thus $a \geq 0$ since we only consider the angle between two gradients in the range of 0 to $\pi$).

Then, we obtain the quadratic expansion of $\mathcal{L}$ as:

$$\mathcal{L}(\theta^+) \leq \mathcal{L}(\theta) + \nabla\mathcal{L}(\theta)^T(\theta^+ - \theta) + \frac{1}{2}\nabla^2\mathcal{L}(\theta)\|\theta^+ - \theta\|^2 \tag{6}$$

and utilize the assumption that $\nabla\mathcal{L}$ is Lipschitz continuous with constant L, we have:

$$\mathcal{L}(\theta^+) \leq \mathcal{L}(\theta) + \nabla\mathcal{L}(\theta)^T(\theta^+ - \theta) + \frac{1}{2}L\|\theta^+ - \theta\|^2 \tag{7}$$

---

[7]Notice that in reality $a^2$ will almost always be smaller than 1 and thus it is sufficient to assume $t < \frac{1}{L}$.

Thus, we plug in the update rule of GradVac to obtain:

$$\mathcal{L}(\theta^+) \leq \mathcal{L}(\theta) - t \cdot \mathbf{g}^T(\mathbf{g} + a\frac{\|\mathbf{g_2}\|}{\|\mathbf{g_1}\|}\mathbf{g}_1 + a\frac{\|\mathbf{g_1}\|}{\|\mathbf{g_2}\|}\mathbf{g}_2) + \frac{1}{2}Lt^2\|\mathbf{g} + a\frac{\|\mathbf{g_2}\|}{\|\mathbf{g_1}\|}\mathbf{g}_1 + a\frac{\|\mathbf{g_1}\|}{\|\mathbf{g_2}\|}\mathbf{g}_2\|^2$$

(Plug in $\mathbf{g} = \mathbf{g}_1 + \mathbf{g}_2$ and re-arrange terms)

$$= \mathcal{L}(\theta) - (t - \frac{1+a^2}{2}Lt^2 + a\phi_{12}(t - Lt^2))(\|\mathbf{g}_1\|^2 + \|\mathbf{g}_2\|^2)$$
$$- (2a(t - Lt^2) + \phi_{12}(2t - Lt^2(1+a^2)))(\|\mathbf{g_1}\| \cdot \|\mathbf{g_2}\|)$$
$$= \mathcal{L}(\theta) - (t - \frac{1+a^2}{2}Lt^2)(\|\mathbf{g}_1\|^2 + \|\mathbf{g}_2\|^2) - 2\phi_{12}(t - \frac{1+a^2}{2}Lt^2)(\|\mathbf{g_1}\| \cdot \|\mathbf{g_2}\|)$$
$$- (a\phi_{12}(t - Lt^2))(\|\mathbf{g}_1\|^2 + \|\mathbf{g}_2\|^2) - (2a(t - Lt^2))(\|\mathbf{g_1}\| \cdot \|\mathbf{g_2}\|)$$

(Remove non-positive terms)

$$\leq \mathcal{L}(\theta) - (t - \frac{1+a^2}{2}Lt^2)(\|\mathbf{g}_1\|^2 + \|\mathbf{g}_2\|^2) - 2\phi_{12}(t - \frac{1+a^2}{2}Lt^2)(\|\mathbf{g_1}\| \cdot \|\mathbf{g_2}\|)$$
$$= \mathcal{L}(\theta) - (t - \frac{1+a^2}{2}Lt^2)(\|\mathbf{g}_1\|^2 + \|\mathbf{g}_2\|^2 + 2\phi_{12}\|\mathbf{g_1}\| \cdot \|\mathbf{g_2}\|)$$
$$= \mathcal{L}(\theta) - (t - \frac{1+a^2}{2}Lt^2)(\|\mathbf{g}_1\|^2 + \|\mathbf{g}_2\|^2 + 2\mathbf{g_1} \cdot \mathbf{g_2})$$
$$= \mathcal{L}(\theta) - (t - \frac{1+a^2}{2}Lt^2)\|\mathbf{g_1} + \mathbf{g_2}\|^2$$
$$= \mathcal{L}(\theta) - (t - \frac{1+a^2}{2}Lt^2)\|\mathbf{g}\|^2$$

The last line implies that if we choose learning rate $t$ to be small enough $t < \frac{2}{L(1+a^2)}$, we have that $t - \frac{1+a^2}{2}Lt^2 > 0$ and thus $\mathcal{L}(\theta^+) < \mathcal{L}(\theta)$ (unless the gradient has zero norm). This tells us applying update rule of GradVac can reach the optimal value $\mathcal{L}(\theta^*)$ since the objective function strictly decreases. $\square$

|                     | en-fr | en-cs | en-hi | en-tr | avg   |
|---------------------|-------|-------|-------|-------|-------|
| GradVac w. HRL_only | 39.07 | 21.51 | 14.92 | 19.63 | 23.78 |
| GradVac w. LRL_only | 39.27 | 21.67 | 14.88 | 19.73 | 23.89 |
| GradVac w. all_task | 38.85 | 21.47 | 14.48 | 19.75 | 23.64 |

Table 6: Comparing which tasks to be included for GradVac. Parameter granularity fixed at all_layer while $\beta$=1e-2.

|                        | en-fr | en-cs | en-hi | en-tr | avg   |
|------------------------|-------|-------|-------|-------|-------|
| GradVac w. whole_model | 38.76 | 21.32 | 14.22 | 18.89 | 23.30 |
| GradVac w. enc_dec     | 39.05 | 21.73 | 14.54 | 19.33 | 23.66 |
| GradVac w. all_layer   | 39.27 | 21.67 | 14.88 | 19.73 | 23.89 |
| GradVac w. all_matrix  | 38.95 | 21.56 | 14.57 | 19.01 | 23.52 |

Table 7: Comparing parameter granularity for GradVac. GradVac tasks fixed at LRL_only while $\beta$=1e-2.

|                            | en-fr | en-cs | en-hi | en-tr | avg   |
|----------------------------|-------|-------|-------|-------|-------|
| GradVac w. $\beta$=1e-1    | 38.72 | 20.74 | 14.52 | 19.25 | 23.31 |
| GradVac w. $\beta$=1e-2    | 39.27 | 21.67 | 14.88 | 19.73 | 23.89 |
| GradVac w. $\beta$=1e-3    | 38.85 | 20.96 | 14.85 | 19.68 | 23.59 |

Table 8: Comparing EMA decay rate $\beta$ for GradVac. Parameter granularity fixed at all_layer and GradVac tasks fixed at LRL_only.

## F  HYPER-PARAMETER SETTINGS

Here, we show how we choose the best hyper-parameter setting for our method. As discussed in Appendix E, there are three hyper-parameter settings for our implementation: (1) which tasks to be considered for GradVac, (2) which layers to measure EMA and perform GradVac, (3) EMA decay rate. Due to the scale of our model on the larger dataset, we use the smaller scale WMT dataset to find the optimal setting and transfer to other experiments. We do this by grid search using average perplexity on the validation set. Below, we demonstrate part of our results for each hyper-parameter to choose from.

First, we examine the effect of what tasks to include for GradVac, i.e. $\mathcal{G}$ in Algorithm 1. We consider three options: (1) HRL_only: only perform GradVac on high-resource languages, (2) LRL_only: only perform GradVac on low-resource languages, (3) all_task: perform GradVac on all languages. Results are shown in Table 6. We find that only conducting GradVac on a subset of languages obtain better performance while it is the best to conduct GradVac on low-resource language only. This is probably because the effective batch sizes of low-resource languages are usually smaller due to the sampling strategy.

Next, we compare the effect of parameter granularity on model quality. This corresponds to setting different model components for GradVac ($\mathcal{M}$ in Algorithm 1). We consider four possibilities, from coarse to fine-grained: (1) whole_model: only perform GradVac once on the entire model, (2) enc_dec: perform separately for encoder and decoder, (3) all_layer: perform individually for each layer in encoder and decoder, (4) all_matrix: perform for each parameter matrix in the model. As shown in Table 7, we find that choosing proper parameter granularity is important, as neither too coarse nor too fine-grained perform the best. This is consistent with our observation made in Section 2. However, we note that our settings are based on NLP tasks and Transformer networks, and therefore the best overall setting for problems of other domains may vary.

Finally, we study how sensitive our method is on the hyper-parameter $\beta$, i.e. the EMA decay rate. Results in Table 8 illustrate that setting an effective "window" of 100 training steps work best for our problem setups. This is expected, as setting a larger $\beta$ value corresponds to conduct GradVac

| | ar | bg | de | en | es | fr | hi | hu | mr | ta | te | vi | avg |
|---|---|---|---|---|---|---|---|---|---|---|---|---|---|
| mBERT | 84.2 | 94.7 | 92.7 | 91.0 | 93.8 | 93.3 | 88.0 | 91.9 | 83.3 | 80.3 | 90.4 | 79.2 | 88.6 |
| + GradNorm | 83.5 | 94.7 | 92.3 | 91.0 | 93.6 | 93.2 | 88.2 | 91.4 | 83.0 | **80.5** | 90.6 | 79.1 | 88.4 |
| + MGDA | **84.4** | 94.5 | 92.3 | 90.4 | 93.5 | 92.7 | 88.1 | 92.3 | 83.4 | **80.5** | 90.2 | 78.7 | 88.4 |
| + PCGrad | 83.7 | 94.8 | 92.6 | 91.5 | 94.2 | 92.8 | **88.5** | 91.7 | **83.7** | **80.5** | 90.8 | 79.4 | 88.7 |
| + GradVac | 84.1 | **95.0** | **93.6** | **91.7** | **94.4** | **93.9** | **88.5** | **92.4** | 83.5 | 79.8 | **90.9** | **79.5** | **88.9** |

Table 9: F1 on the POS tasks of the XTREME benchmark.

more aggressively, and vice versa. In general, we find our best settings to be consistent across tasks in this paper.

# G  XTREME EXPERIMENTS

## G.1  FINETUNING DETAILS

We also conduct experiments on the XTREME benchmark (Hu et al., 2020) for cross-lingual transfer tasks. While other work mostly focus on zero-shot cross-lingual transfer (finetune on English training data and then evaluate on the target language test data), we use a different setup of multi-task learning such that we finetune multiple languages jointly and evaluate on all languages. Notice that our goal is not to compare with state-of-the-art results on this benchmark but rather to examine the effectiveness of our proposed method on pre-trained multilingual language models. We therefore only consider tasks that contain training data for all languages: named entity recognition (NER) and part-of-speech tagging (POS).

The NER task is from the WikiAnn (Pan et al., 2017) dataset, which is built automatically from Wikipedia. A linear layer with softmax classifier is added on top of pretrained models to predict the label for each word based on its first subword. We report the F1 score. Similar to NER, POS is also a sequence labelling task but with a focus on synthetic knowledge. In particular, the dataset we used is from the Universal Dependencies treebanks (Nivre et al., 2018). Task-specific layers are the same as in NER and we report F1. We select 12 languages for each task randomly.

We use the multilingual BERT (Devlin et al., 2018) as our base model, which is a Transformer model pretrained on the Wikipedias of 104 languages using masked language modelling (MLM). It contains 12 layers and 178M parameters. Following Hu et al. (2020), we finetune the model for 10 epochs for NER and POS, and search the following hyperparameters: batch size {16, 32}; learning rate {2e-5, 3e-5, 5e-5}.

## G.2  POS RESULT

We evaluate all multi-task baselines on the POS tasks in Table 9. We find that our proposed method outperforms other methods on average, consistent with results in other settings (Section 4).

