# OpenReview forum: "Gradient Vaccine: Investigating and Improving Multi-task Optimization in Massively Multilingual Models"
_ICLR.cc/2021/Conference — ICLR 2021 Spotlight_

### Official Review · AnonReviewer2 · 2020-10-26
**An extension of existing approach, benefits not clear**

**Rating:** 6
**Confidence:** 3

**Review:**

The paper studies the behaviour of gradient similarities across languages in multilingual NMT models. They find gradient similarities mirror language similarity. Hence, they look at method to gradient-based methods for multilingual NMT. They apply PCGrad to the multilingual NMT task and also extend this method to address the cases when the task gradients have only weak similarity.

**Strengths**
- I find the analysis of gradients and language similarities interesting and adds to the understanding of multilingual NMT models.
- The GradVac method addresses the case of positive cosine values and thus allowing for faster mitigation of interference while training.
- The experimentation and analysis are strong. Particularly, the study of hyper-parameters to discover the best settings is very useful.

**Weaknesses**
- The proposed method is only a minor extension of the PCGrad method and the novelty is limited.
- The gains against corresponding PCGrad variant is very limited - probably won't be statistically significant. So, it is not clear if there is a major benefit to the GradVac procedure over PCGrad. A statistical significance test of the BLEU scores using standard bootstrap sampling will be useful.

**Questions for the authors**
- Did you explore setting gradient objectives based on language similarity?

---

> ### Author Response · Authors · 2020-11-22
> **Responses to Reviewer 2**
>
> Thank you very much for your comprehensive review and valuable feedback. We address your comments one by one as following:
>
> [Extension over PCGrad]
> We agree that our proposed method is an extension of PCGrad. However, the derived GradVac is more generic and flexible, which enables its application on more challenging real-world settings such as MNMT. In particular, one of  our main contributions is the systematic study of characteristics of the multi-task optimization procedure for multilingual models, which reveals that multilingual tasks exhibit diverse yet (mostly) positive gradient similarities. Because of these two properties, PCGrad is not effective for multilingual models. Specifically, as discussed in Section 3, PCGrad implicitly sets a universal gradient similarity objective of 0 for all task pairs. This is verified in both Figure 3 and Table 1, where most gradients are non-conflicting and thus PCGrad only obtains marginal gains over the naive multilingual baseline. On the other hand, GradVac addresses this key limitation and is thus applicable for more problem setups. Therefore, we believe that our proposed method is simple yet important.
>
> [Performance gain]
> Thank you for your advice! We have performed significant tests and the gains of GradVac are statistically significant with p < 0.05 for all results except hi-en in Table 1. As a matter of fact, our results are averaged across 5 random runs and the standard deviations are around 0.15 for results in Table 1. Therefore, it is clear that GradVac consistently improves over PCGrad and other baselines, as it outperforms PCGrad by more than 1 average BLEU score on Eng-to-Any settings and 0.5 on Any-to-Eng settings. We will include these results and make this clearer in an updated version.
>
> [Setting objectives based on language similarity]
> Our analysis in Section 2 reveals that gradient similarities reflect language proximities, and thus the proposed method has already used objectives based on language similarities. We didn’t further explore setting them manually because: (1) it is hard to set accurate values (e.g. should we set the same values between En-Fr and En-Es or not?) (2) as shown in Figure 4(c), gradient similarities evolve over the training process and thus setting constant values can be suboptimal (as verified by one of our ablation studies in Table 1 row 7).

---

### Official Review · AnonReviewer1 · 2020-10-28
**A good paper**

**Rating:** 7
**Confidence:** 4

**Review:**

The paper proposes a novel method, GradientVaccine, to improve multi-task optimization on a massive multilingual translation and named entity recognition model. They investigate the loss function geometry on many language pairs and use the idea to encourage more geometrical parameter updates. This approach extends Gradient Surgery (a.k.a PCGrad) by adding an exponential moving average with a term $\phi^{(t)}_{ijk}$ and $\beta$ to address the limitation of PCGrad. I can say this approach is straightforward but very useful. The presented idea is very elegant and has been convinced by the theoretical foundation.

Strengths:
- The paper proposes a multi-task training method by calculating the gradient similarity and used them as a trajectory of the optimization.
- The method is beneficial in massive multilingual NMT and XTREME benchmarks.

Weaknesses:
- The results of the proposed method seems consistent. However, the authors may only run the experiment once. It would be great if the authors can also provide a significant test of their results. And interestingly, in POS, the method is not as effective as NMT and XTREME.

I have some questions:
- How does $\beta$ affect the gradient similarity objective? Can you also provide an ablation study on this?

Overall, I enjoy reading this paper, and the experimental results are strong, and the paper is solid. The authors compare this method with other important baselines. In summary, this is a good paper. It presents a straightforward and useful idea for multi-task learning.

**Post-rebuttal**

> I want to thank the author for addressing my concerns. Overall, this is an exciting paper with comprehensive ablation studies and analysis. It provides an effective multi-task training method for multilingual tasks. The authors also support the method with a strong mathematical foundation. Thus, I would like to keep my positive rating.

---

> ### Author Response · Authors · 2020-11-22
> **Responses to Reviewer 1**
>
> Thank you very much for your comprehensive review and valuable feedback. We address your comments one by one as following:
>
> [Results]
> Thank you for pointing this out! In fact, our reported results are averaged across 5 random runs in all experiments except for those in Table 2 (due to computational cost consideration). For example, the standard deviation is around 0.15 for BLEU scores in Table 1, so it is clear that GradVac consistently improves the performance. In addition, we conduct significant tests as suggested and report that the gains are statistically significant with p < 0.05 for all results except hi-en in Table 1. We will add additional significant tests and make this clear in the final version.
>
> [Ablation study on $\beta$]
> We conduct ablation studies on all parameter settings in Appendix F. Specifically, the results of using different $\beta$ values are shown in Table 8. Our results show that setting values that are too small or too large can be suboptimal. This is because smaller $\beta$ values tend to induce more stable gradient similarity objectives, and thus very small $\beta$ fail to effectively adapt to gradient geometries evolved across training stages, while very large values are too sensitive to noise.

---

### Official Review · AnonReviewer3 · 2020-10-28
**Review comments for paper 2546**

**Rating:** 6
**Confidence:** 3

**Review:**

This paper conducts comprehensive analyses and a method to the multi-task training in multilingual models. By analyzing the gradient similarity of two tasks in multilingual NMT, this paper reveals that gradient similarities reflect language proximities, correlate with model quality, and also evolve with layers and training steps. Furthermore, this paper proposes a method called GradVac to improve the multi-task training over standard monolithic training. Experiment results show GradVac achieves better accuracy than other multi-task optimization methods.

The paper is well-written and easy to follow. The motivation is clear, the analyses on the gradient similarity are interesting, and method proposed is effective according to experiments comparison.  While I like the investigation on multilingual model in Section 2, I have some questions on the proposed method in Section 3:
1) How do you choose the task to alter the gradient? By random? For example, we always change g1 (by random) for any two gradients g1 and g2. If g1 in current batch is estimated more accurate than g2 or g1 can contribute more on the model optimization than g2, do we still alter g1 while leave g2 unchanged?  How about first choose a not good gradient than alter it instead of alter by random?
2) Does it harm the optimization of a task if we change its gradient? How can we ensure that the benefits brought by solving the conflicts of two gradients is worthy compared to the drawbacks brought by altering the gradient of a task?
3) How about consider more tasks at a time instead of only 2 tasks? Can the proposed method be extended smoothly? Since it is more practical as there are multiple tasks trained at the same time.
4) In equation 3, with EMA, the gradient similarity objective may be more stable than the computed gradient similarity at each step. What is the intuition behand? I do not see GradVac is more preemptively in this sense.

I expect the author can answer the points above, and then I can adjust my final score.

---

> ### Author Response · Authors · 2020-11-22
> **Responses to Reviewer 3**
>
> Thank you very much for your comprehensive review and valuable feedback. We address your comments one by one as following:
>
> [Which tasks to apply GradVac]
> In our proposed method, we control which task gradients to manipulate. This is better seen from Algorithm 1 in Appendix E, where we only alter gradients for tasks in the set $\mathcal{G}$. We experiment with different $\mathcal{G}$ in Appendix F. In particular, results in Table 6 show that it is the best to alter gradients for low-resource languages. This is consistent with your intuition that less reliable gradients should be prioritized for manipulation, since high-resource languages tend to induce more robust gradients within a batch (due to the sampling strategy and their abundance of data). We will make this more explicit in an updated version.
>
> [Trade-off between single-task vs multi-task performance]
> It is possible that altering the gradient of an individual task can hurt its own optimization. However, our overall goal here is to improve the averaged multi-task performance and we have shown the effectiveness of the proposed method both theoretically and empirically. Theoretically, our proof in Appendix E.3 shows that our method is guaranteed to improve the averaged multi-task performance when altering gradients. And this is further verified by our extensive experiments using different settings.
>
> [Working with multiple tasks]
> In fact, our proposed method is indeed designed for multiple tasks as detailed in Algorithm 1. All experiments in Section 4 involve multiple tasks trained in a single model simultaneously (up to 25) as well. When working with many tasks, the method is largely the same except iterating through all tasks in a random order (yet we can still control which gradients to manipulate as discussed above).
>
> [Intuition of EMA and meaning of preemptive]
> The intuition behind using EMA variables is based on our analysis in Section 2, where we find gradient similarities evolve across different training stages. On the other hand, gradients can be noisy in a single batch. So to capture the accurate trend of gradient similarities, we use the more robust EMA variables such that they reflect the gradient objectives of the current training stage. In Table 8 we conduct ablation studies on EMA parameters and in Table 1 we experiment with setting constant gradient objectives. These results highlight the importance of setting appropriate objectives over different training steps.
> Besides, we apologize for the confusion but the word ‘preemptive’ is used to characterize GradVac against PCGrad. Since GradVac can adaptively use more aggressive gradient similarity objectives while PCGrad is not effective for positive gradient similarities, we say that GradVac is more preemptive compared to PCGrad. We will make this clearer in the later version.

---

### Official Review · AnonReviewer4 · 2020-10-28
**This work takes aim at an interesting problem of optimizing multilingual neural machine translation (MNMT) model. Although MNMT is inherently a multi-task modeling approach, less emphasis have been given on achieving an optimal performance on all of the tasks involved. The proposed approach (GradVac) takes into account similarity between tasks and demonstrates better performance can be achieved by focusing on parameter updates that are geometrically aligned.**

**Rating:** 8
**Confidence:** 4

**Review:**

Summary
Taking multilingual NMT (MNMT) into account, this work, investigates better model optimization alternative, that is in part can be attributed as a multi-task optimization problem. MNMT's are quite beneficial from different perspectives (improving low-resource languages, efficiency, etc). However, their inherently multi-task nature requires more focus on how to gist out the best possible learning for each of the languages pairs. With a potential impact on the optimization of other multi-task models, this work asks how model the similarity between model gradients is crucial in multi-task settings, and how to best optimize MNMT models focusing on the typologically similarity of languages. By analyzing the geometry of the NMT model objective function, authors indicate that computing similarity along gradient provides information on the relationship between languages and the overall model performance. Authors argue the analysis of the gradient helps to identify the point of limitation in multi-task learning, which the work aims to address, by focusing the parameter updates for tasks that are similar or close in terms of geometrical alignment (also known as Gradient Vaccine /GradVac/).

Experimental results are provided from multilingual tasks involving 10^9 magnitude model training examples and several languages pairs. Mathematical proof and theoretical details of the proposed optimization approach GradVac are detailed in comparison with previous approach (such as Gradient Surgery). Experimental results shows the proposed GradVac to contribute for the improvement of model performance. These findings underline the importance of taking into account language proximity for a better optimization approach and model improvements in general.



Pros / Reason for the Score
After my assessment of the proposed approach and the visible advantage of GradVac, I am voting for an accept score. Below my points of the pros and cons of this work. It's my hope authors will address the cons and the questions raised in the rebuttal period.

- This work raises an important question of optimization in a multi-task model, particularly for multilingual NMT models where an optimization approach is quite rare and recent progress in MNMT mainly focuses on improving performance. Hence the findings in this work, can provide further insight on how to best optimize an MNMT model and potentially set a new standard training mechanism for future works in MNMT.

- From the experimental results, particularly its quite interesting to see how the proposed approach (GradVac) improves the high-resource languages (on the left side of Figure 6 (b)). I think in massive MNMT models while there is huge gain (naturally) for low-resource cases, the high-resource pairs tends to degrade. This work shows an interesting mechanism to address performance degradation for certain pairs in an MNMT model and to maintain an improvement trend  for all of the language pairs involved.


Cons and Questions
- Considering the language similarity, this work focused on typologically similarity (that deals with the characteristics of the language structure), is there any consideration for genetic similarity, or any other similarity measure between languages the authors considered? Or why is the typological similarity the primary/only choice for this work?

- As in Yu et al. 2020, where the PCGrad approach is used to project the gradient of task i to the plane of task j, was there any motivation behind not to adapt or asses this approach in MNMT before going / do the authors have any comment why this approach does lag behind from the GradVacc.? - Perhaps this is related to the assumption PCGrad is not fit a positive gradient similarities - a case in this work?

- One of the advantages of MNMT model is efficiency (as also mentioned in this work), however, when we deal with model training or even inference the paper does not mention the complexity that can be introduced by the application of GradVac, can the author provide the details on this?

- Page (P) 1: mentions one of the motivations for the work is to investigate ways to optimize the single language-agnostic objective for training an MNMT model that leverages training data of multiple pairs - if this work is aiming at optimizing based on task relatedness - did it consider for instance training MNMT models that are language family specific and see how that relatedness correlates with the approach in this work and the baseline MNMT models (such as Any->En or En->Any)?

- What is the impact of training only two MNMT models Any->En and En->Any, why not Any<>Any? Wouldn't this make a lot more sense from the point of having multiple tasks (in terms of observing different language characteristics both at the encoder and decoder side of the model)?
Similarly, the Any<>Any that is employed (shown in Figure 2.) gradient similarities correlates positively with model quality. In other words authors clearly demonstrated the En>Any direction gradient similarity is quite low with respect to Any>En, in my understanding using Any<>Any model throughout the experiment makes more sense by constructing a real multi-tasking MNMT model, where we can also see the proposed approaches effectiveness.

- Not sure if I am missing it, if it correct that we do not have comparison of the proposed optimization approaches with other optimizations from the results in Figure 6? At least with PCGrad ?


Comments
- In an ideal case, I would go for evaluating a multilingual model that is not English centric to properly construct a real multilingual model. I understand the experimental design here, specifically this is the data (En<>Any) in general or available in-house for the authors. Yet, with recent progresses in multilingual NMT and zero-shot NMT approaches, its becomes realistic now to incrementally augment data for the non English pairs (can leverage monolingual data of the Any languages too), hence, resulting in more pairs. Such a setting of Any-Any can even further reflect how the optimization is beneficial.

- please re-arrange the figures, I see discussion about Figure 5 while there is Figure 3 and 4 beforehand - if possible.

---

> ### Author Response · Authors · 2020-11-22
> **Responses to Reviewer 4**
>
> Thank you very much for your comprehensive review and valuable feedback. We address your comments one by one as following:
>
> [Language similarity]
> We agree that studying the effect of different choices of similarity measures is an interesting research question for future exploration and we will add a discussion in our final version. We choose typological similarity mainly because it is informative and popular. Prior works have analyzed typological measurement extensively [1,2,3] and found it beneficial for understanding and improving NLP models [4,5]. For instance, even phylogenetically related languages can still exhibit divergences in typology so it is more informative for modelling similar linguistic structures.
>
> [Inefficacy of PCGrad]
> Your intuition is correct that PCGrad is not effective for the case of positive gradient similarities, which is a common case for many NLP tasks including MNMT (since languages are similar semantically and/or synthetically). As we stated in Section 3.2, PCGrad implicitly sets a universal gradient similarity objective of zero for all tasks, hindering its applicability in problem settings where tasks exhibit diverse inter-task relatedness including MNMT.
>
> [Efficiency]
> Thank you for pointing this out! For memory efficiency, the resulting models will have the same numbers of parameters for deploying as typical MNMT models and thus enjoy the same benefits. For computational efficiency, the proposed method will have the same order of complexity with the original multi-task training paradigm. In practice, on our large-scale models, GradVac will have the identical inference speed and a slow down in training speed of 20%. However, please notice that our implementation is based on Tensorflow v1 without heavy optimization, which uses a static computational graph such that all redundant nodes will be calculated. We will add a discussion in the paper.
>
> [Language-family-specific models]
> We agree that language-family-specific models are interesting to study. However, they are not very effective in improving performance on low-resource language pairs, an important motivation for MNMT models. This is mainly because languages within the same language family tend to have similar amounts of data (e.g. we have more data for most European languages but less for most African languages). In addition, there are so many ways to cluster languages based on any prior, so we choose to leave them for more extensive ablations in a future work.
>
> [Any-to-Any Model]
> We definitely agree that studying the Any-to-Any model would be the ideal setup. However, we choose to use Any-to-En and En-to-Any models for two reasons: (1) The data that we had are English-centric, as is the case for many research published in this area. (2) For the purpose of controlled experiments, it is easier to analyze Any-to-En and En-to-Any models separately. For example, it is hard to interpret the gradient similarity measured between En->Fr and Es->En (Is it due to difference within the encoder (En/Es) or the decoder (Fr/En)). Therefore, training separate models allows us to remove such confounding factors and understand the optimization behaviors better. That said, although out of the scope of this paper, we believe that training a true Any-to-Any will become more practical with new techniques/data and should be addressed in future research. We will add a discussion in the final version.
>
> [Baseline in Figure 6]
> The computational resources required to train on our massive datasets prohibit us from experimenting with all possible settings. In our preliminary experiments, as well as results in Table 1, we found that baselines including PCGrad fail to improve MNMT performance robustly and thus we refrained from training them on a full scale. We will make this point clearer.
>
> We also thank you for your suggestions on paper formatting and we will be sure to update our manuscript accordingly.
>
> [1] How Good are Typological Distances for Determining Genealogical Relationships among Languages? Rama and Kolachina. COLING 2012.
> [2] Modeling Language Variation and Universals: A Survey on Typological Linguistics for Natural Language Processing. Ponti et al., 2018.
> [3] Bridging Linguistic Typology and Multilingual Machine Translation with Multi-View Language Representation. Oncevay et al., EMNLP 2020.
> [4] Investigating multilingual nmt representations at scale. Kudugunta et al., EMNLP 2019.
> [5] Choosing Transfer Languages for Cross-Lingual Learning. Lin et at., ACL 2019.

---

### Decision · Program_Chairs · 2021-01-07
**Final Decision**

**Decision:**

Accept (Spotlight)

**Comment:**

This paper proposes a scalable optimization method for multi-task learning in multilingual models.

Pros:
1) Addresses a problem which has not been explored much in the past
2) Presents very good analysis to show the limitations of existing methods.
3) Good results.
4) Well written

Cons:
1) Some missing details about various choices made in the experiments (mostly addressed in the rebuttal)

This is a very interesting and useful work and I recommend that it should be accepted.